# Expression of antenatal symptoms of common mental disorders in The Gambia and the UK: a cross-sectional comparison study

Katie Rose M Sanfilippo [1], Vivette Glover,[2] Victoria Cornelius,[3] Rita T Amiel Castro,[4] Bonnie McConnell,[5] Buba Darboe,[6] Hajara B Huma,[6,7] Hassoum Ceesay,[7] Paul Ramchandani [8], Ian Cross,[9] Lauren Stewart[10]

**Correspondence to**
Katie Rose M Sanfilippo;
ksan91@gmail.com

## ABSTRACT

**Objectives** It is important to be able to detect symptoms of common mental disorders (CMDs) in pregnant women. However, the expression of these disorders can differ across cultures and depend on the specific scale used. This study aimed to (a) compare Gambian pregnant women's responses to the Edinburgh Postnatal Depression Scale (EPDS) and Self-reporting Questionnaire (SRQ-20) and (b) compare responses to the EPDS in pregnant women in The Gambia and UK.

**Design** This cross-sectional comparison study investigates Gambian EPDS and SRQ-20 scores through correlation between the two scales, score distributions, proportion of women with high levels of symptoms, and descriptive item analysis. Comparisons between the UK and Gambian EPDS scores were made by investigating score distributions, proportion of women with high levels of symptoms, and descriptive item analysis.

**Setting** This study took place in The Gambia, West Africa and London, UK.

**Participants** 221 pregnant women from The Gambia completed both the SRQ-20 and the EPDS; 368 pregnant women from the UK completed the EPDS.

**Results** Gambian participants' EPDS and SRQ-20 scores were significantly moderately correlated ($r_s$=0.6, p<0.001), had different distributions, 54% overall agreement, and different proportions of women identified as having high levels of symptoms (SRQ-20=42% vs EPDS=5% using highest cut-off score). UK participants had higher EPDS scores (M=6.5, 95% CI (6.1 to 6.9)) than Gambian participants (M=4.4, 95% CI (3.9 to 4.9)) (p<0.001, 95% CIs (−3.0 to −1.0), Cliff's delta = −0.3).

**Conclusions** The differences in scores from Gambian pregnant women to the EPDS and SRQ-20 and the different EPDS responses between pregnant women in the UK and The Gambia further emphasise how methods and understanding around measuring perinatal mental health symptoms developed in Western countries need to be applied with care in other cultures.Cite Now

## BACKGROUND

It is important to be able to recognise and treat symptoms of common mental disorders (CMDs), including depression and anxiety,

## STRENGTHS AND LIMITATIONS OF THIS STUDY

⇒ Findings support studies conducted in other parts of Africa, adding to the literature by exploring differences within a new cultural context: the Gambia.

⇒ This study compares two different scales (Edinburgh Postnatal Depression Scale (EPDS) and Self-reporting Questionnaire (SRQ-20)) within the same cultural context in addition to comparing one scale (EPDS) across cultures.

⇒ This study includes a relatively large sample size across The Gambia collected across 10 antenatal clinics in the West Coast and North Bank regions.

⇒ The EPDS and SRQ-20 have not been validated in The Gambia against a clinical gold-standard (clinical interview) to screen for possible common mental disorders.

during pregnancy. These symptoms affect both the mother and can also have long-term adverse effects on her child.[1] Perinatal mental health problems are a particular challenge in low-income and middle-income countries (LMICs) where they can be at least twice as frequent as in higher-income countries and are often associated with stigma.[2] Thus, it is necessary to be able to identify these disorders both for intervention and research purposes. Short, simple self-rating questionnaires are often the easiest way to do this. However, women in different cultures and settings can have different experiences and presentation of CMD symptoms which may lead to under-recognition or misidentification of psychological distress,[3] impacting where women might seek support and what treatment options they are offered, if any. For instance, Oates *et al*,[4] in a qualitative cross-cultural study of post-natal depression, found that while all cultures described a condition of 'morbid unhappiness' after childbirth, not everyone saw it as an illness remediable by health interventions.

Likewise, the presentation of symptoms is not uniform across different populations. Previous works in different African populations have suggested that CMD symptoms are experienced in a more somatic way.[5]

The Edinburgh Postnatal Depression Scale (EPDS), a commonly used tool to measure CMD symptoms, deliberately minimises items of a somatic nature since it has been argued that physical changes in pregnancy might confound interpretation of CMD symptoms.[6] Even so, it has been recommended specifically for perinatal depression detection in LMICs.[7] However, Cox *et al*[6] note that the use of the EPDS in certain cultures may be inappropriate. Given that some populations experience CMD symptoms in a more somatic way, the use of a scale which precludes their detection is likely to underestimate the incidence of CMD symptoms in such populations. An alternative to the EPDS is the Self-reporting Questionnaire (SRQ-20),[8] developed by the WHO for the detection of symptoms of CMDs in LMICs. It is suggested to be useful for measuring perinatal CMDs in sub-Saharan Africa due to its inclusion of somatic symptom items.[9 10]

While the somatisation of CMDs has been documented in a number of African populations (e.g., in Etheopia, Malawi and Nigeria[5 10 11]) the existing bias for research studies to concentrate on western populations, as well as the diversity of the African continent, motivates exploring this question in different populations and with different methodological approaches. In the current study, we compare how pregnant women of Gambian nationality (living in The Gambia) scored on two common measures of CMD symptoms, the EPDS and the SRQ-20. We also compare CMD symptoms, as measured by the EPDS, of these Gambian women against those of UK nationals (living in London, UK). Therefore, the objectives of this study were to explore differences between two scales within the same cultural context (The Gambia) and one scale across two different contexts (The UK and The Gambia).

## METHODS

This study is a cross-sectional comparison study where we use a comparative and descriptive analysis approach. This is not a validation study but rather exploratory and descriptive in nature to compare how two different scales used to measure perinatal CMD symptoms perform within The Gambia and, also, how the same scale (EPDS) performs across two different countries. For this paper, the Strengthening the Reporting of Observational Studies in Epidemiology (STROBE) guidelines were followed, and the checklist can be found in online supplemental material 1.

### Participants and setting

A convenience sample of 221 Gambian pregnant women between 10 and 30 weeks gestation (145=Mandinka speaking and 76=Wolof speaking) were assessed from May to August 2018. This was a sufficient sample size as it

was determined that a sample size of 194 was needed to detect a correlation coefficient of 0.2 and a sample size of 124 was needed for a cross-sectional study design to detect a prevalence of 26% (the estimate of prevalence of antenatal depression for women in LMICs[12]) when using a two-sided correlation test, 5% significance level test ($\alpha=0.05$) with 80% power ($\beta=0.2$).[13 14]

The Gambia is a country on the West African coast and is made up of eight different ethnic groups all with their own languages and culture. Islam is the predominant religion in The Gambia and is a powerful factor in shaping life, beliefs and expectations. Participants were recruited from 10 different antenatal clinics across the western part of The Gambia (for more information, see online supplemental material 2). Any woman who attended the clinic, was above 18 years old and spoke either Mandinka or Wolof was approached. Women were excluded if they had previous or current psychosis or a history of a late term miscarriage.

For the second part of the study, existing data from a convenience sample of 368 pregnant women living in the UK from Queen Charlotte's and Chelsea Hospital in London was used. These data were part of a previous longitudinal study where women were recruited from April 2013 to April 2014. Exclusion criteria was therefore different from that of the Gambian cohort and excluded participants who did not speak and/or write English, did not have a device with internet access, had a multiple pregnancy, in vitro fertilisation, severe medical problems, pregnancy medical problems (including abnormal fetus) or severe psychiatric problems (eg, psychosis, suicidality or drug addiction).

### Measurement scales

The EPDS[15] is a 10-item scale that was designed to measure postnatal depression and anxiety symptoms that has subsequently been validated for antenatal use.[16] It has been used with perinatal populations across Africa[17] and in perinatal populations in The Gambia,[18–21] though a validated version in Mandinka or Wolof could not be obtained. Participants answer on a four-point Likert-scale (from 0 to 3) how often they have experienced a specific symptom within the last week. The three anxiety items (items 3, 4 and 5) have been shown to form a valid anxiety subscale (EPDS-3[22]). A higher total score (out of a total of 30) corresponds to a higher level of symptoms.

The SRQ-20[8] is a 20-item scale that measures symptoms of CMDs. It was developed to be used in primary care settings in LMICs and has also been used in perinatal populations across Africa[9 10 23] and The Gambia[21] though, in The Gambia, it has not been validated against a gold-standard clinical interview. The SRQ-20 includes items measuring common somatic symptoms associated with anxiety and depression (headaches, low appetite, poor digestion and sleep problems) (items 1–3, 5, 7 and 18–20) as well as other psychological and physical/somatic symptoms (feeling frightened, unhappy, worthless and low-energy) (items 4, 6, 8 and 9–17).[8 10] To each

item, participants answer either yes=1 or no=0. The scores range from 0 to 20 with a higher score indicating higher levels of CMD symptoms.

Both scales were translated into Mandinka and Wolof using methods based on recommendations from the WHO,[24] Hanlon *et al*[25] and Cox *et al*.[6] First, the scales were translated into Mandinka and Wolof by bilingual experts. The terms used and concepts around mental distress were informed by a large qualitative study held prior to this work. An expert panel, consisting of experts in Gambian culture and language and an expert in mental health from the Ministry of Health and Social Welfare, held a discussion and then refined the translation. Both were then back-translated into English and reviewed again by the expert panel that developed the final version.

## Procedure

Local midwives in The Gambia identified possible participants. If participants met the inclusion criteria, they were given information about the study by the two research assistants (RAs). Both RAs were trained psychiatric nurses and one was also a midwife. Verbal informed consent was obtained, witnessed and recorded via RA signature or participant thumbprint. Due to low levels of literacy in The Gambia,[26] participants' EPDS and SRQ-20 responses were collected orally by the RAs in alternating order per participant. There were no order effects found when using an independent samples t-test (p>0.05, 95% CIs (4.37 to 4.30)). The RAs were not blinded to which scale they were administering. Oral administration of self-report tools involves an interaction between the questionnaire, the respondent and the interviewer which can lead to potential biases.[27] These areas of potential bias were mitigated through the careful training and monitoring of the data collection method used throughout the study. In the previous study conducted in the UK, eligible participants were invited to take part by the researcher and written informed consent was obtained. EPDS scores were measured online at about 21 weeks gestation.

## Statistical analysis

All statistical analysis was run using R.[28] To compare the demographic and pregnancy characteristics independent samples t-tests were used. To investigate differences and similarities in the EPDS and the SRQ-20 within the Gambian sample, a Spearman's correlation test was conducted, differences in the distributional properties were examined, and the average score of the EPDS-3 subscale and its contribution to the total EPDS score of the sample was computed. Individual item frequencies on the SRQ-20 (# of yes responses out of total responses) were also calculated. To explore the impact of the somatic SRQ-20 items, the average score of the combined somatic items and their contribution to the total SRQ-20 score was calculated. Previous research has validated optimum cut-off scores for identifying perinatal CMDs within different populations in Africa.[7 17] The scores of 6, 7 and 8 were chosen as cut-offs for the SRQ-20, based on previous

validation research in Ethiopia and Malawi.[9 10] Proportions of specific agreement and kappa statistics based on the identification of participants below or above these different cut offs were calculated based on methods explained by Cicchetti and Feinstein[29] and then Kappas were calculated.

To investigate differences across countries using the EPDS, differences in the distributional properties were explored. Three cut-off scores, 9, 10 and 12, were chosen for the EPDS based on previous research in the UK[6] and across Africa.[17] The average score of each item and of the EPDS-3 subscale was calculated. Its contribution to the total EPDS score of the sample was also computed. Differences between the total EPDS score between the UK and The Gambia were calculated using a Mann-Whitney U test.

## Patient and public involvement

We engaged with patients and the public and relevant stakeholders in The Gambia to inform the research approach. This included how to best administer the questionnaires, the best way to recruit participants and potential cultural considerations which need to be made. To do this, we held a half day meeting in The Gambia with delegates from the Ministry of Health and Social Welfare, the National Centre for Arts and Culture, a local obstetrician, patient advocate groups and students in Public Health and Psychiatric Nursing. In general, all stakeholders felt this was a worthy and important endeavour as maternal mental health is a relatively unexplored area in The Gambia. One crucial point which was discussed was how we would sensitively manage and appropriately signpost women who report any concerning symptoms, such as suicidal ideation or domestic violence. It was decided that if this were to happen, the woman would receive in the moment front line counselling with the RAs who are both trained psychiatric nurses. Then, if needed, she would be referred on to the Community Mental Health Team for further management. For women who report domestic violence emergencies and cases that require immediate intervention, the RAs would connect the woman with the Gender-Based Violence focal person. For other cases, the RAs would refer the woman to the One Stop Centre at Serekunda General Hospital or Edward Francis Small Teaching Hospital.

## RESULTS
### Demographic information

Table 1 shows the demographic characteristics of the Gambian and UK samples. UK participants were significantly older (p<0.001, 95% CIs (5.4 to 7.2), d=1.4) and had fewer children (p<0.001, 95% CIs (0.6 to 1.2), d=0.6) and pregnancies (p<0.001, 95% CIs (2.0 to 3.6), d=0.9) than the Gambian participants.

### Comparing the EPDS and the SRQ-20 in the Gambian sample

The overall total scores for both scales were significantly moderately correlated $r_s$=0.6, p<0.001 (see online

**Table 1** Demographic information for Gambian and UK samples

| | Gambian sample (n=221) M (SD) | UK sample (n=314) M (SD) |
|---|---|---|
| Age* | 27.3 (5.9) | 33.4 (4.3) |
| GA | 22.9 (5.5) | 21.1 (1.4) |
| Parity* | 2.3 (2.1) | 1.5 (0.7) |
| Gravida* | 3.5 (2.3) | 2.0 (2.0) |
| **Gambian sample** | **Total n (% of 221)** | |
| Marital status | n (% of 221) | |
| Single/divorced/widowed | 6 (3%) | |
| Married (monogamous) | 160 (72%) | |
| Married (polygamous) | 55 (25%) | |
| Education Level | n (% of 221) | |
| None | 13 (6%) | |
| Informal (arabic) | 101 (46%) | |
| Primary | 29 (13%) | |
| Secondary/tertiary | 78 (35%) | |
| **Occupation** | **n (% of 220)** | |
| Housewife | 138 (62%) | |
| Other | 82 (37%) | |
| Husband's occupation | n (% of 219) | |
| Skilled work | 97 (44%) | |
| Manual/trade work | 122 (55%) | |
| **UK sample** | **Total sample of 368†** | |
| **Marital status** | **n (% out of 314)** | |
| Married/civil partnership | 233 (74%) | |
| Living with partner | 64 (20%) | |
| Single | 11 (4%) | |
| Other | 6 (2%) | |
| **Education level** | **n (% out of 315)** | |
| Left before GCSE/O-levels | 2 (1%) | |
| GCSE/O-levels | 16 (5%) | |
| A-levels | 23 (7%) | |
| Vocational training | 13 (4%) | |
| University degree | 159 (51%) | |
| Higher degree | 102 (32%) | |
| **Occupation** | **n (% out of 314)** | |
| Professional | 197 (63%) | |
| Managerial | 46 (15%) | |
| Skilled non-manual | 16 (5%) | |
| Skilled manual | 13 (4%) | |
| Partly skilled | 10 (3%) | |

**Table 1** Continued

| | Gambian sample (n=221) M (SD) | UK sample (n=314) M (SD) |
|---|---|---|
| Not applicable | 22 (7%) | |
| Other | 10 (3%) | |
| **Ethnic group** | **n (% out of 314)** | |
| Caucasian | 209 (67%) | |
| Indian/Pakistani/Bangladeshi | 21 (7%) | |
| Middle Eastern | 12 (4%) | |
| Afro/Afro-Caribbean | 12 (4%) | |
| South American/Hispanic | 4 (1%) | |
| Far Eastern | 3 (1%) | |
| Mixed | 12 (4%) | |
| Other | 41 (13%) | |

All categories were informed by the categorisations used by the Ministry of Health and Social Welfare (MoHSW) in The Gambia.
*Indicates a significant difference (p<0.05) between the groups using independent samples t-tests.
†The total sample in the previously conducted study was 368 but there was missing demographic data. There was also missing demographic data for some of the categories for Gambian participants. The total used for each category is included in the table.
GA, gestational age; GCSE, General Certificate of Secondary Education. ; Primip, Primiparious. Housewife includes women who stated they did not have a job ('none').

supplemental material 3). Figure 1 compares the distributions of the total EPDS and SRQ-20 scores and the proportion of women above different cut-off scores.

There was low agreement between the two scales in the Gambian sample in identifying those above the various cut-offs scores chosen (kappa's ranging from 0.05 to 0.25). The average positive agreement—when both scales detected a participant above the cut-off—was low (28%). Negative agreement—when both identified the participant as below the cut-off—was higher (67%). The average overall agreement—positive and negative agreement combined—was 54%. Online supplemental material 4 displays the proportion of participants identified using different cut-off scores, proportions of specific agreement and kappa's between the SRQ-20 and the EPDS.

Table 2 shows the responses of the Gambian sample to the individual items on the SRQ-20. The somatic symptom items had a higher average score (M=4.0, SD=2.0) than psychological/cognitive/functional symptom items (M=3.0, SD=2.5). The somatic symptom items contributed 63% to the total SRQ-20 score. The psychological/cognitive/functional symptom items contributed 37%.

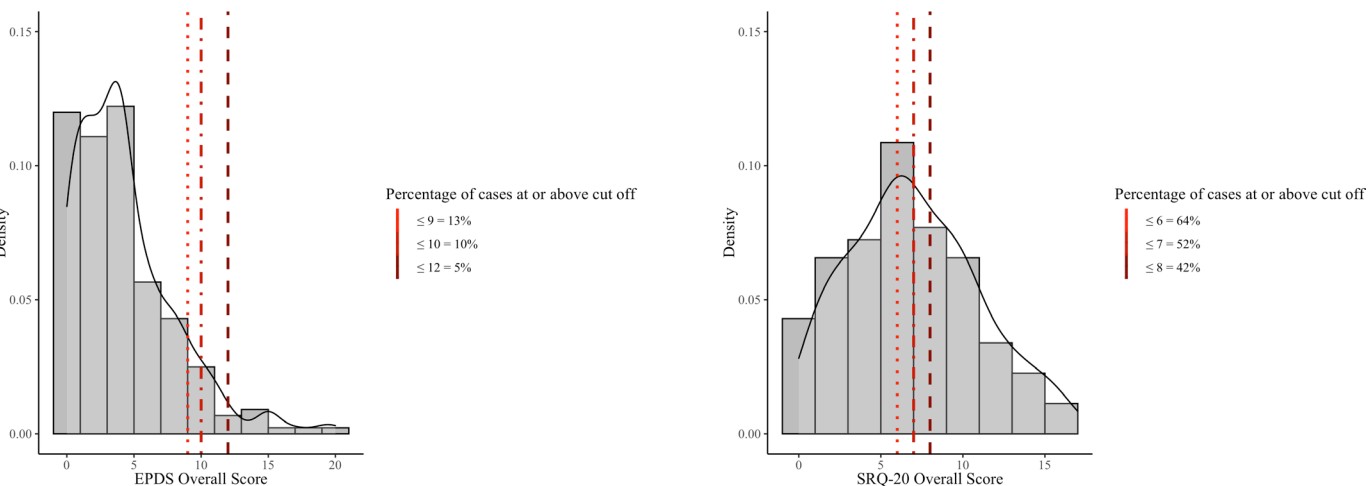

**Figure 1** Histogram of different cut-off scores for EPDS and SRQ-20. EPDS, Edinburgh Postnatal Depression Scale; SRQ-20, Self-reporting Questionnaire.

### Comparing EPDS scores in the Gambia and the UK

The UK average EPDS score (M=6.5, SD=4.4, 95% CI (6.1 to 6.9)) was significantly higher than the Gambian average score (M=4.4, SD=3.7, 95% CI (3.9 to 4.9) (p<0.001, 95% CIs (−3.0 to −1.0), Cliff's delta=−0.3). A Mann-Whitney U Test was used because the UK scores were slightly positively skewed (skewness=0.63), while the Gambian EPDS scores were highly positively skewed (skewness=1.37), reflecting a higher frequency of lower scores in the Gambian sample. Figure 2 shows the distribution of total EPDS scores in the Gambian and UK samples as well as the proportion above the chosen cut-off scores.

Of note, 13% of UK participant's EPDS scores were ≥12 compared with 5% in the Gambian sample. With a cut-off of ≥10, 27% of UK versus 10% of Gambian participants were identified and when using the cut-off of ≥9, 34% UK versus 13% Gambian participants were identified (figure 2). Thus, with each cut-off, two to three times more women scored above the selected cut-off score from the UK sample compared with the Gambian sample.

Table 3 shows how the two samples scored on the individual items on the EPDS. In the Gambian sample, the

item scores were generally lower. The Gambian sample had a significantly higher average score (M=0.4, SD=0.6) compared with the UK sample (M=0.3, SD=0.5) only on the reverse coded item 2 (I have looked forward with enjoyment to things) (p<0.004, 95% CIs (0.3 to 0.1), d=0.3). For all other significant differences, the UK sample had a higher item average (see table 3). The EPDS includes an anxiety subscale (EPDS-3) consisting of items 3, 4 and 5 [22 30] This subscale average was also significantly higher for the UK sample (M=3.5, SD=2.0) compared with the Gambia sample (M=1.7, SD=1.7) (p<0.004, 95% CIs (1.5 to 2.1), d=0.9). The anxiety subscale scores contributed 54% to the total UK EPDS score. For the Gambian sample, the subscale contributed to 39% of the total EPDS score. Two of the most endorsed items in The Gambian sample were within the anxiety subscale (items 4 and 5).

### DISCUSSION

When comparing scores on the EPDS between the Gambian and UK samples, the UK participants had significantly higher total EPDS scores. 13% of UK

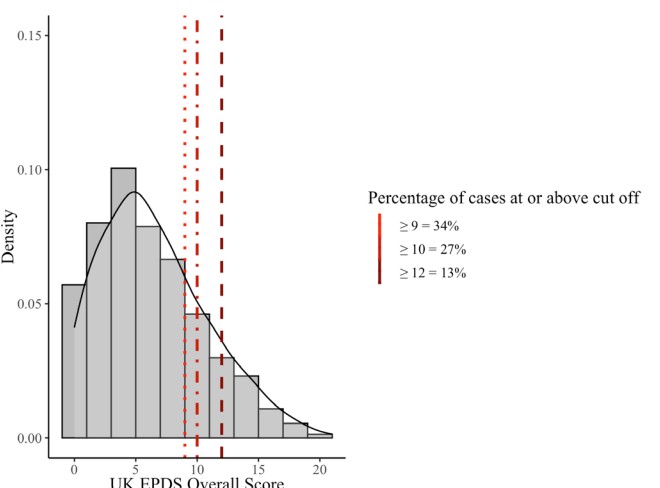
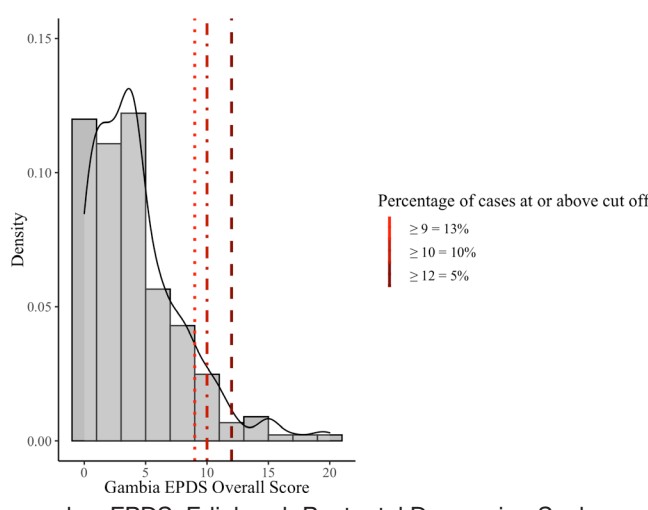

**Figure 2** EPDS distributions and cut-off score for UK and Gambia samples. EPDS, Edinburgh Postnatal Depression Scale.

**Table 2** SRQ-20 item, item category frequency

| SRQ-20 item | | Yes N (% out of 221) |
|---|---|---|
| 1 | Do you often have headaches? | 167 (76) |
| 2 | Is your appetite poor? | 121 (55) |
| 3 | Do you sleep badly? | 93 (42) |
| 4 | Are you easily frightened? | 82 (37) |
| 5 | Do your hands shake? | 24 (11) |
| 6 | Do you feel nervous, tense or worried? | 89 (40) |
| 7 | Is your digestion poor? | 78 (35) |
| 8 | Do you have trouble thinking clearly? | 58 (26) |
| 9 | Do you feel unhappy? | 94 (43) |
| 10 | Do you cry more than usual? | 20 (9) |
| 11 | Do you find it difficult to enjoy your daily activities? | 45 (20) |
| 12 | Do you find it difficult to make decisions? | 74 (33) |
| 13 | Is your daily work suffering? | 80 (36) |
| 14 | Are you unable to play a useful part in life? | 60 (27) |
| 15 | Have you lost interest in things? | 74 (33) |
| 16 | Do you feel that you are a worthless person? | 37 (17) |
| 17 | Has the thought of ending your life been on your mind? | 10 (5) |
| 18 | Do you feel tired all the time? | 116 (52) |
| 19 | Do you have uncomfortable feelings in your stomach? | 121 (55) |
| 20 | Are you easily tired? | 109 (49) |
| **Item category** | | **M (SD)** |
| 1–3, 5, 7, 18–20 | *Somatic symptoms | 4.0 (2.0) |
| 4, 6, 8, 9–17 | *Psychological/cognitive/functional symptoms | 3.0 (2.5) |

*Categorisation of items based on Stewart *et al* 2009.[10]
SRQ-20, Self-reporting Questionnaire.

participants' scores met the threshold for high levels of symptoms (≥12) compared with only 5% of the Gambian participants. This could, of course, be taken at face value to indicate potentially higher levels of CMS symptoms in the UK population. However, the comparison of EPDS and SRQ-20 scores reveals an alternative view. While both scales were significantly moderately correlated ($r_s$=0.6, p<0.001), they had different distributions and only 54% overall agreement. Additionally, when using previously validated cut-off scores from other contexts, only 5% of Gambian participants met the threshold for high levels of symptoms with the EPDS, while that number rose to 42% when the SRQ-20 was used.

The aim of this comparative cut-off score analysis was to investigate how various cut-off scores validated in other perinatal contexts would compare in our sample. As has been shown in similar work around the world, the two scales generally perform quite differently within the same context and the same scale performs quite differently across two different contexts. It may be that the EPDS significantly underestimates the incidence of CMD symptoms in this Gambian sample due to how it is administered, the response format, and cultural differences in the presentation of perinatal CMD symptoms as has been found in other similar perinatal populations (eg., Shrestha *et al* and Bluett-Duncan[31 32]). It may also be that the EPDS does not capture the somatic way in which CMD symptoms are experienced. However, without a clinical interview to compare against, it is difficult to know the reason for the observed differences.

Somatic symptoms, rather than those of depressive mood, were more frequently endorsed by our Gambian sample. This may be in part because of differences of language. Different ways of describing the world can lead speakers of different languages to have different ways of thinking about the world.[33] For example, item 2 (I have looked forward with enjoyment to things) was the only item where the average score of the Gambian sample was significantly higher than the average score of the UK sample. It may be that the idea of anticipating or expecting enjoyment in your future might go against ideas of humility and patience and faith in God (any plans for the future are normally concluded with 'inshallah', God willing).

Items related to sadness or crying were some of the least endorsed by the Gambian participants. Differences in CMD symptom presentation when measured using translated self-report measures may not reflect Gambian indigenous understandings of symptom causation and, in fact, these indigenous understandings may contrast with biomedical perspectives.[34] For example, a systematic review found that local language versions of the EPDS in LMICs had lower precision for identifying true cases of perinatal CMDs compared with the original English version.[31] Additionally, Molenaar *et al*[35] found that in a perinatal population in rural Ethiopia, most women recognised the existence of perinatal mental distress states, but did not call such distress a discrete illness, such as depression. Instead, these mental distress states were generally seen as non-pathological reactions to difficult circumstances.

Some researchers have raised the concern that measuring somatic CMD symptoms in populations with high levels of physical disease might mistakenly attribute these symptoms to a depressive syndrome.[15] In a perinatal population, this may present an even more difficult problem as somatic symptoms related to pregnancy are expected to be experienced and change throughout this time. However, an important study by Stewart *et al*[10] found that the inclusion of somatic items did not affect the test performance of the SRQ-20 in their Malawi postnatal population as validated by a semistructured diagnostic interview for depressive disorder. This current study is in

**Table 3** EPDS item and subscale averages for the UK and Gambia samples

| | | UK sample | Gambia sample |
|---|---|---|---|
| Item # | | M (SD) | M (SD) |
| 1 | I have been able to laugh and see the funny side of things | 0.2 (0.5) | 0.3 (0.5) |
| 2 | I have looked forward with enjoyment to things* | 0.3 (0.5) | 0.4 (0.6) |
| 3 | I have blamed myself unnecessarily when things went wrong* | 1.2 (0.8) | 0.4 (0.7) |
| 4 | I have been anxious or worried for no good reason* | 1.4 (0.9) | 0.7 (0.8) |
| 5 | I have felt scared or panicky for no good reason* | 0.9 (0.8) | 0.6 (0.9) |
| 6 | Things have been getting on top of me* | 1.0 (0.8) | 0.6 (0.8) |
| 7 | I have been so unhappy that I have had difficulty in sleeping | 0.4 (0.7) | 0.5 (0.8) |
| 8 | I have felt sad or miserable* | 0.6 (0.8) | 0.5 (0.7) |
| 9 | I have been so unhappy that I have been crying | 0.5 (0.6) | 0.3 (0.7) |
| 10 | The thought of harming myself had occurred to me | 0.1 (0.3) | 0.1 (0.3) |
| | **Anxiety subscale (3, 4, 5)***<br>**All other items** | 3.5 (2.0)<br>2.7 (2.6) | 1.7 (1.7)<br>3.0 (2.9) |
| | **Total score*** | **6.5 (4.4)** | **4.4 (3.7)** |

Bonferroni adjusted alpha level of 0.004 was used per test (0.05/13).
*Indicates a significant difference when using an independent samples t-test at p<0.004.
EPDS, Edinburgh Postnatal Depression Scale.

agreement with previous research that used the SRQ-20 in other perinatal African populations and found the somatic items from the SRQ-20 were, indeed, the most frequently endorsed.[9 10]

Previous research has also found that somatisation of mental health disorders in general, and perinatal disorders in particular, is common in many African populations.[5 9 25 36] This supports the idea that in many global majority cultures, the symptoms of dysphoric disorders seem to be mostly somatic and different from the Western systems.[36] Taken together, as has been found in previous studies,[3 37 38] our results further emphasise how methods and understanding around measuring perinatal mental health symptoms developed in Western countries need to be employed with care when working within and across different cultures.

### LIMITATIONS
While careful consideration and translation of the SRQ-20 and EPDS helped to ensure semantic validity of the scales for use in The Gambia, the scales have not been validated against a clinical gold-standard (clinical interview) to screen for possible CMD, highlighting a limitation of this work. Other studies conducted in sub-Saharan African countries with perinatal populations studies have shown good criterion validity of the SRQ-20 and the EPDS against the clinical interview.[10 25 39–41] However, validation against a clinical interview was difficult in The Gambia partly because there are few trained professionals (only two trained psychiatrists for the whole of the country) able to conduct these interviews in the two local languages. This limitation highlights an important consideration needed when conducting validation studies in low resource settings with few local trained health professionals. Even though the scales were not validated, we feel the comparison of the scales' performances when using possible cut-off scores is an interesting and helpful exploratory process.

For the purposes of this comparative analysis, we combined the scores from the Mandinka and Wolof languages. Mandinka is a Mande language and Wolof is classified as a Niger-Congo language and there is a mutual influence between them. If validation were to be done in the future, ideally each of the languages would be assessed separately.

How the EPDS was administered was different in the two settings. The EPDS was self-completed in written form in the UK versus delivered and responded to orally in The Gambia due to the low literacy rate in the population. Other factors include differences in the recruitment process, purpose and setting (hospital vs community), and in demographic factors such as age and parity. These differences were partly because existing data from a previously conducted study in England was used. These other factors might explain, in part, some of the differences reported on the EPDS between participant groups. However, it seems unlikely that these factors alone could explain the major differences in response.

The SRQ-20's binomial response format, compared with the EPDS likert-scale response format, may have influenced the differences observed in the performance of these two tools. The binomial response format is more easily administered and understood by respondents with low literacy.[38] For example, a study comparing these two scales in a Ghanaian pregnant population found that it was important to alter the format of the EPDS in Ghana,

for it to be understandable to women.[42] Responses like 'hardly ever' and 'sometimes' might have been difficult to distinguish when administered orally.

## CONCLUSION

In conclusion, these findings support studies conducted with perinatal populations in other parts of sub-Saharan Africa.[9 10 25 43] This study is able to add to this growing body of work by exploring differences within a new cultural context. While caution should be taken when generalising these results, this study is one of the first to explore the expression of antenatal mental health symptoms in The Gambia when using two common CMD measurement tools. It will be of interest in the future to explore further the cultural reasons for the discrepancies found.

Crucially, this study helps to further underline the importance of investigating differences in scale performance within and across cultures to ensure that the measurement scale chosen is sensitive to cultural differences in the understanding and presentation of perinatal CMDs. Without careful exploration of these differences, research in this area may have detrimental consequences in detection and therefore treatment of perinatal CMDs in LMICs.

**Author affiliations**
¹Centre for Healthcare Innovation Research, City University of London, London, UK
²Institute of Reproductive and Developmental Biology, Imperial College London, London, UK
³Imperial Clinical Trials Unit, School of Public Health, Imperial College London, London, UK
⁴Psychologisches Institut, Universitat Zurich, Zurich, Switzerland
⁵School of Music, Australian National University, Canberra, Australian Capital Territory, Australia
⁶Ministry of Health and Social Welfare The Gambia, Banjul, Gambia
⁷National Centre for Arts and Culture The Gambia, Banjul, Gambia
⁸Faculty of Education, University of Cambridge, Cambridge, UK
⁹Centre for Music & Science, Faculty of Music, Cambridge University, Cambridge, UK
¹⁰Psychology Department, Goldsmiths University of London, London, UK

**Acknowledgements** We would like to thank the women and clinicians in the antenatal clinics in the Gambia and at Queen Charlotte's and Chelsea Hospital in the UK. Thank you also to Pa Bakary Sonko and Charlotte Hanlon for their advice and support, Giles Partington for analysis advice and Jane Offerman for helping with administration.

**Contributors** The manuscript was prepared by KRMS and all authors approved the final draft. KRMS and BM helped with data management and data collection materials. BD provided local knowledge, expertise in health provision and healthcare in The Gambia and organisation in the field liaising with antenatal clinics. HC provided management of the funds and transportation in The Gambia. HBH helped complete all data collection and input and provided knowledge of mental health and reproductive health in the Gambia. KRMS completed data cleaning and analysis. VC provided methodological and statistical expertise. VG and PR provided expertise in perinatal mental health. IC provided expertise on culture and language and BM, HC and BD provided expertise on Gambian culture and language. RC was responsible for all data collection and management for the UK sample. All authors, except VG, RC and IC, spent time in The Gambia supervising the project on the ground. LS is the guarantor, grant holder and principal investigator and oversaw all aspects of the study.

**Funding** This study was funded by the MRC-AHRC Global Public Health: Partnership Awards scheme (MR/R024618/1) awarded to Professor Lauren Stewart. This writing of this manuscript was supported by a South East Network for Social Sciences/The Economic and Social Research Council funded postdoctoral fellowship awarded to Dr Katie Rose Sanfilippo while at Goldsmiths, University of London (Grant Reference Number: ES/V010158/1).

**Competing interests** None declared.

**Patient and public involvement** Patients and/or the public were involved in the design, or conduct, or reporting or dissemination plans of this research. Refer to the Methods section for further details.

**Patient consent for publication** Not required.

**Ethics approval** This study involves human participants and was approved by Goldsmiths University Ethics Committee (1392), The Gambia Government/MRC Gambia joint Ethics Committee (R018 015V2) and the Australian National University Ethics Committee (2018/235). Participants gave informed consent to participate in the study before taking part.

**Provenance and peer review** Not commissioned; externally peer reviewed.

**Data availability statement** Deidentified participant data that correspond to the results reported in this article are available upon reasonable request from the corresponding author (katie-rose.sanfilippo@city.ac.uk).

**ORCID iDs**
Katie Rose M Sanfilippo http://orcid.org/0000-0003-2236-3307
Paul Ramchandani http://orcid.org/0000-0003-3646-2410

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
