## [Reviewer comments · BMJ Open]

ARTICLE DETAILS

TITLE (PROVISIONAL)	Expression of Antenatal Symptoms of Common Mental Disorders in The Gambia and the UK: A cross-sectional correlation comparison study
AUTHORS	Sanfilippo, Katie Rose; Glover, Vivette; Cornelius, Victoria; Castro, Rita; McConnell, Bonnie; Darboe, Buba; Huma, Hajara; Ceesay, Hassoum; Ramchandani, Paul; Cross, Ian; Stewart, Lauren

VERSION 1 – REVIEW

REVIEWER	Fellmeth, Gracia University of Oxford Nuffield Department of Population Health
REVIEW RETURNED	10-Aug-2022

GENERAL COMMENTS	Thank you for the opportunity to review this manuscript which I very much enjoyed reading. The study addresses the important topic of perinatal CMD symptoms across two different cultures, highlighting the often overlooked challenges of using screening tools which have been developed in 'Western' settings in other parts of the world. I have listed below some minor comments for the authors' consideration, but overall I think this is an important and well-conducted study that will be of wide interest. - In the Background section, I wondered about the use of the term 'bodily' symptoms and whether 'somatic' or 'physical' symptoms might be more fitting / more widely-used? I leave it to your discretion as you may know better than me - it's just I've not come across the former term before. - Participants: there are more exclusion criteria listed for the UK participants than the Gambian participants. I was wondering why different criteria applied - was it because for the English sample data from a previous study was used? Either way I would perhaps add an explanatory sentence in the methods and/or discussion section to clarify this. - I was pleased to see that the WHO guidelines around the translation of tools was followed. However, I wondered if the translated versions of the EPDS and SRQ were formally validated against a clinical interview prior to this study? This would be a crucial step in establishing the validity of the tools and determining an appropriate cut-off for the Gambian setting. I would recommend a more detailed discussion about this in the Limitations section. Currently it is mentioned as a limitation but only briefly but I think it warrants more attention. - Minor point but were the EPDS and SRQ administered independently / blindly by the two RAs? Or did the same RA
--

	administer both tools? It would be helpful if you could report this in the methods. I'd also be interested to know if the order of administration had any effect on scores. - It is great to hear about the PPI activities. Please could you provide more information on the outcomes of your discussions with patients and stakeholders? Did you make any changes to your research design in light of the PPI discussions? - I would recommend avoiding the term 'case' in the results section. (I can only see it once or twice in the first paragraphs, but perhaps check if any more instances of it). It's not only because the cut-offs have not been validated against a gold standard in The Gambia, but also more generally because even if validated, screening tools don't tell you 'caseness' (only symptoms indicative of a disorder). Most of the time you refer to individuals scoring above/below the cut-off which I think is fine. - The results are very interesting and provide wonderful insights into the differences between the two populations. I find the exploration of individual item endorsement and comparison of total scores and means particularly fascinating. My only slight concern is with the section comparing the proportion of women who score above a given threshold. This to me becomes problematic because it assumes that the same given cut-off is appropriate across both settings, which is unlikely to be true. Without validation against a gold standard it's not possible to say what the most appropriate cut-off is in The Gambia. But if for example validation had been conducted and a cut-off of 8 yielded the highest sensitivity/specificity, while a cut-off of 10 was best in the UK - then I would argue that it would be more appropriate to compare these two respective cut-offs to each other, rather than comparing the same cut-off across two different contexts? I hope this makes sense. I'm just not sure how helpful this particular analysis is. Perhaps if you're keen to keep this section you could leave it as it is, but pick up on some of the issues (i.e. that it may not be meaningful to compare the same (unvalidated) cut-off across two settings) in the discussion? I also would not choose this result to be the first thing to mention in the Discussion - you have so many novel and exciting and more meaningful results you could start with!) Discussion - Personally I think you're likely to be right that the EPDS is under-estimating CMD symptoms. However, in the absence of a gold standard comparison I would be wary about making this assumption so explicitly. I would suggest re-word so that it's suggested as a possibility, whilst being clear that without a clinical interview to compare against it's not possible to know whether the EPDS or the SRQ is providing the more accurate measure. - I wondered whether you considered the differences between the two languages in the Gambian setting. Are the two languages very similar to each other? Purists would argue that each language (even if just a dialect of each other) should be validated separately, and different cut-offs might potentially apply to each language. I don't feel that you need to change anything in your analyses, but I think it warrants mentioning in the Discussion section that ideally each of the languages would have been assessed separately.
--	---

REVIEWER	Rahman, M University of Malaysia Sarawak, Community Medicine and Public Health
REVIEW RETURNED	08-Jan-2023

GENERAL COMMENTS	I already mentioned my comments in the text - The reviewer provided a marked copy with additional comments. Please contact the publisher for full details.
---

REVIEWER	Rucci, Paola Alma Mater Studiorum – University of Bologna, Department of Biomedical and Neuromotor Sciences
REVIEW RETURNED	24-Jan-2023

GENERAL COMMENTS	The paper by Sanfilippo et al. aims at comparing the expression of antenatal symptoms of Common Mental Disorders between pregnant women in Gambia and the UK. Women from Gambia completed both the SRQ-20 (Self Reporting Questionnaire) and the EPDS (Edinburgh Postnatal Depression Scale), while women from the UK completed only the EPDS. As a general comment, the paper is interesting and helps to underline the importance of investigating differences in antenatal symptoms within and across different cultures. However, there are some methodological issues to clarify and the results need to be improved. In particular, the EPDS is an established and validated scale. I would be inclined to suggest to limit the analyses to the comparisons of the total score, the subscale (anxiety and depression scores) and the items between samples, in line with the aims of the study. Exploratory factor analysis in the Gambian sample leads to an unstable solution with three factors and one item with a cross-loading and is not a useful add-on to the paper. The Bland-Altman analysis comparing the EPDS and the SRQ-20 scales in the Gambian sample is not appropriate because the two scales have a different content. A correlation between the two scales would offer more useful information to the readers. Specific comments. Page 3 of 46 Line 43. Authors should report mean EPDS scores along with 95%CIs. Page 7 of 46 Line 125, 126. The items that comprise the anxiety subscale are reported inconsistently in the paper. They mention items 1,2, 4 and subsequently (line 246) they mention items 3, 4 and 5. Please check. Line 132-134. The items comprising somatic symptoms and those comprising psychological symptoms should be listed in the text and in table 2. Page 8 of 46 Line 156. In 'Statistical analysis', the statistical tests used to compare the demographic and pregnancy characteristics and the
---

item frequencies are not indicated. The tests used and the corresponding significance level should also be reported in Table 1

Page 9 of 46

Line 171, 172. Please remove the sentence "The average scorecomputed" because it is a repetition of the same sentence reported in lines 159, 160.

Page 10 of 46

Line 195. In Table 1, the absolute and percentage frequencies should be reported for the total samples of each country, indicating the missing data for each variable. Apparently for UK data, demographic information is available only for 314 women.

Line 195. In Table 1, a parenthesis is missing for GA's standard deviation.

Line 195. Please round off percentages, means and standard deviations to one decimal figure throughout the text and tables.

Page 12 of 46

Line 207-209. Given the different content of the scales, the Bland-Altman analysis is useless. I would suggest reporting only the Spearman's correlation coefficient between the EPDS and the SRQ-20.

Line 218-221. The authors do not report which and how many items comprise the subscales. However, it seems that subscales consist of a different number of items. If this is the case, comparing the means of subscales is not useful. The authors should consider standardizing the scores before comparison.

Page 13.

Lines 226-230.

Please use Mann-Whitney test to compare mean EPDS scores between countries, given the skewness in the Gambian sample.

Page 14 of 46

Line 234. I would suggest comparing the proportion exceeding the 3 cut-off scores in the two countries using chi-square test and reporting the results in a table.

Line 249. The percentage of contribution of the anxiety subscale to the total UK EPDS score is reported as 60%. However, if we divide 3.49 (the average scores of items #3, #4 and #5) by 6.51 (the total score), the percentage should be 54%. Please check.

Line 254. Please report in the table 3 the subscale average scores.

Page 15 of 46

Line 256. The Bonferroni correction to the probability level applied to 10 items should be 0.05/10 and not 0.05/13. Please revise.

Line 264-266. The variance explained by each factor should be reported at the beginning of table 4 for the two countries.

Page 23.

Please replace ref [19] with Kabir K, Sheeder J, Kelly LS. Identifying postpartum depression: are 3 questions as good as 10? Pediatrics. 2008 Sep;122(3):e696-702, where the three items comprising the anxiety scale are listed.

Page 27 of 46

	Figure 2. Authors should specify in the x-axis of the histogram "Gambian EPDS overall score"
--	--

REVIEWER	Pawlby, Susan Institute of Psychiatry, Psychology & Neuroscience at King's College London
REVIEW RETURNED	29-Jan-2023

GENERAL COMMENTS	This manuscript is extremely well written and addresses some of problems faced in identifying common mental health problems (CMDs) in Lower and Middle Income Countries when using screening instruments developed in English for use in Higher Income Countries. In this study the authors firstly compared Gambian pregnant women's responses to the Edinburgh Postnatal Depression Scale (EPDS), developed in the UK, and the Self-reporting Questionnaire (SRQ-20) developed by the WHO for use across cultures. Secondly they compared responses to the EPDS in pregnant women in The Gambia and UK. Both instruments (EPDS and SRQ-20), were rigorously translated, back-translated and given orally to the Gambian women, with each tool contributing to the understanding of how CMDs present in pregnant women in The Gambia. As in other African countries there was more endorsement of somatic symptoms, suggesting that the SRQ might work better in identifying CMDs among pregnant women in these cultures than the EPDS. Comparison of the Gambian women's scores on the EPDS with those from a group of UK women confirmed that women in the UK scored significantly higher than the Gambian women and that there was a different factor structure. Although no structured diagnostic interview was given to the Gambian women (a limitation acknowledged by the authors), other African studies have shown good criterion validity of the SRQ against the SCID and the MINI. This study in The Gambia is important as it complements other African studies examining ways in which CMDs can be assessed in antenatal clinics so that pregnant women can be supported through the perinatal period. The reliance on screening instruments and indeed interviews that have originally been formulated in Western countries and often in the English language, some of them many years ago, still poses a problem, despite the extremely rigorous attention that has been paid to translation, using consensus rating and focus groups. Few studies have used instruments developed specifically for use in a given cultural setting. More qualitative research conducted within country by local health professionals with pregnant women should help characterise perinatal CMDs in different African contexts. 2
--

	Following the concluding sentence that 'This study helps to further underline the importance of investigating differences in scale performance within and across cultures to ensure that the measurement scale chosen is sensitive to cultural differences in the understanding and presentation of perinatal common mental disorders', the authors may like to refer to a recent study by Bluett-Duncan who presented another way of overcoming the problems of crosscultural research while using multi-item scales of mental health. He explored Differential Item Functioning (DIF) comparing a group of Indian mothers with a group of UK mothers in their responses to the items of the EPDS. In his thesis he describes the development of a set of anchoring vignettes (AVs) in order to assess the impact of DIF on the EPDS. His findings showed that the mothers in the two cultural settings rated the items on the EPDS very differently (as they did in The Gambia compared with the UK). Prior to Differential Item Functioning (DIF) adjustment, rates of depression at 12 months were significantly higher in UK participants, compared with Indian participants, but after DIF adjustment, rates were significantly higher in Indian participants. Differences in response style indicated that Indian women were under-reporting depressive symptoms when compared with the UK women and showed that scores unadjusted for DIF should not be taken at face value as they may mask the extent of maternal mental ill-health in LMICs.
--	--

VERSION 1 – AUTHOR RESPONSE

Reviewer: 1

Dr. Gracia Fellmeth, University of Oxford Nuffield Department of Population Health

Comments to the Author:

Thank you for the opportunity to review this manuscript which I very much enjoyed reading. The study addresses the important topic of perinatal CMD symptoms across two different cultures, highlighting the often overlooked challenges of using screening tools which have been developed in 'Western' settings in other parts of the world. I have listed below some minor comments for the authors' consideration, but overall I think this is an important and well-conducted study that will be of wide interest.

- 1. In the Background section, I wondered about the use of the term 'bodily' symptoms and whether 'somatic' or 'physical' symptoms might be more fitting / more widely-used? I leave it to your discretion as you may know better than me - it's just I've not come across the former term before.*

We would like this paper to be as accessible to as wide an audience as possible and aim to be consistent in our terminology. Therefore, as per your suggestion, we have changed bodily to somatic throughout. You can see these as track changes in the document.

2. *Participants: there are more exclusion criteria listed for the UK participants than the Gambian participants. I was wondering why different criteria applied - was it because for the English sample data from a previous study was used? Either way I would perhaps add an explanatory sentence in the methods and/or discussion section to clarify this.*

We have now clarified within the method section that the English (UK) sample data was from a previous study with different aims and objectives. This is why the exclusion criteria is different.

For the second part of the study, existing data from a convenience sample of 368 pregnant women living in the UK from Queen Charlotte's and Chelsea Hospital in London was used. This data was part of a previous longitudinal study where women were recruited from April 2013 to April 2014. Exclusion criteria was therefore different to that of the Gambian cohort and excluded participants who did not speak and/or write English, did not have a device with internet access, had a multiple pregnancy, in vitro fertilization, severe medical problems, pregnancy medical problems (including abnormal foetus), or severe psychiatric problems (e.g. psychosis, suicidality or drug addiction).

(Participants and Setting, Page 6)

We also added more detail in the limitations section:

How the EPDS was administered was different in the two settings. The EPDS was self-completed in written form in the UK versus delivered and responded to orally in The Gambia due to the low literacy rate in the population. Other factors include differences in the recruitment process, purpose, and setting (hospital versus community), and in demographic factors such as age and parity. These differences were partly because existing data from a previously conducted study in England was used.

(Limitations, Page 19)

3. *I was pleased to see that the WHO guidelines around the translation of tools was followed. However, I wondered if the translated versions of the EPDS and SRQ were formally validated against a clinical interview prior to this study? This would be a crucial step in establishing the validity of the tools and determining an appropriate cut-off for the Gambian setting. I would recommend a more detailed discussion about this in the Limitations section. Currently it is mentioned as a limitation but only briefly but I think it warrants more attention.*

We were not able to validate the translated versions against clinical interview. Part of the reason for this was the lack of trained health professionals within the country who would be able to administer this interview in Mandinka and Wolof. There are only two trained psychiatrists in The Gambia for its

population of about 2 million people. We agree this is an important (as well as interesting) limitation to the study.

Therefore, we have added more discussion around this point within the limitation section of this paper.

While careful consideration and translation of the SRQ-20 and EPDS helped to ensure semantic validity of the scales for use in The Gambia, the scales have not been validated against a clinical gold-standard (clinical interview) to screen for possible CMD, highlighting a limitation of this work. Other studies conducted in sub-Saharan African countries with perinatal populations studies have shown good criterion validity of the SRQ-20 and the EPDS against the clinical interview [10,25,38–40]. However, validation against a clinical interview was difficult in The Gambia partly because there are few trained professionals (only two trained psychiatrists for the whole of the country) able to conduct these interviews in the two local languages. This limitation highlights an important consideration needed when conducting validation studies in low resource settings with few local trained health professionals. Even though the scales were not validated we feel the comparison of the scales' performances when using possible cut-off scores is an interesting and helpful exploratory process.

(Limitations, Page 18)

- 4. Minor point but were the EPDS and SRQ administered independently / blindly by the two RAs? Or did the same RA administer both tools? It would be helpful if you could report this in the methods. I'd also be interested to know if the order of administration had any effect on scores.*

In the Gambia, each RA administered both tools to each participant. They knew which tool they were administering so they were not blinded. We have now added this to the Procedure section.

Each RA did alternate the order in which they administer the tools. We found that there were no order effects for the Gambian sample and this has also been briefly included.

Due to low levels of literacy in The Gambia[26], participants' EPDS and SRQ-20 responses were collected orally by the RAs in alternating order per participant. There were no order effects found when using an independent samples t-test ($p > 0.05$, CIs [4.37, 4.30]). The RAs were not blinded to which scale they were administering.

(Procedure, Page 8)

- 5. It is great to hear about the PPI activities. Please could you provide more information on the outcomes of your discussions with patients and stakeholders? Did you make any changes to your research design in light of the PPI discussions?*

We had a very fruitful discussion before the study commenced. We have added a bit more information in this section about the outcomes of the discussion and what changes we made, specifically around dealing with disclosures of suicidal ideation or domestic violence.

In general, all stakeholders felt this was a worthy and important endeavour as maternal mental health is a relatively unexplored area in The Gambia. One crucial point which was discussed was how we would sensitively manage and appropriately sign post women who report any concerning symptoms, such as suicidal ideation or domestic violence. It was decided that if this were to happen, the woman would receive in the moment front line counselling to talk through these issues with the RAs who are both trained psychiatric nurses. Then, if needed, she would be referred on to the Community Mental Health Team for further management. For women that report domestic violence emergencies and cases that require immediate intervention, the RAs would connect the woman with the Gender Based Violence focal person. For other cases, the RAs would refer the woman to the One Stop Center at Serekunda General Hospital or Edward Francis Small Teaching Hospital. (Patient and Public Involvement section, Page 10)

6. *I would recommend avoiding the term 'case' in the results section. (I can only see it once or twice in the first paragraphs, but perhaps check if any more instances of it). It's not only because the cut-offs have not been validated against a gold standard in The Gambia, but also more generally because even if validated, screening tools don't tell you 'caseness' (only symptoms indicative of a disorder). Most of the time you refer to individuals scoring above/below the cut-off which I think is fine.*

Thank you for catching this. You are right that we did try to avoid using case or caseness for the exact reason you state. We have taken out the use of 'case' in the result section. We have kept the one use of "case" in the introduction as this is a sentence which is summarising the findings from a systematic review which was identifying cases.

There was also low agreement between the two scales in the Gambian sample in identifying those above the various cut-offs scores chosen (kappa's ranging from 0.05-0.25).

(Results, Page 12)

7. *The results are very interesting and provide wonderful insights into the differences between the two populations. I find the exploration of individual item endorsement and comparison of total scores and means particularly fascinating. My only slight concern is with the section comparing the proportion of women who score above a given threshold. This to me becomes problematic because it assumes that the same given cut-off is appropriate across both settings, which is unlikely to be true. Without validation against a gold standard it's not possible to say what the most appropriate cut-off is in The Gambia. But if for example validation had been conducted and a cut-off of 8 yielded the highest sensitivity/specificity, while a cut-off of 10 was best in the UK - then I would argue that it would be more appropriate to compare these two respective cut-offs to each other, rather than comparing the same cut-off across two different contexts? I hope this makes sense. I'm just not sure how helpful this particular analysis is. Perhaps if you're keen to keep this section you could leave it as it is, but*

pick up on some of the issues (i.e. that it may not be meaningful to compare the same (unvalidated) cut-off across two settings) in the discussion? I also would not choose this result to be the first thing to mention in the Discussion - you have so many novel and exciting and more meaningful results you could start with!

Thank you for your helpful suggestion. We agree that unvalidated cut-off score comparison can be misleading and have made sure we address this more clearly in the paper. We chose to do this analysis as an exploratory exercise to look at how different cut-offs scores validated in similar contexts would compare in our sample. As has been shown in similar work around the world the two scales generally perform quite differently within the same context and the same scale performs quite differently across two different contexts. However, what you have pointed out is right – because the scales are not validated in The Gambia we cannot say exactly what may be influencing these observed differences.

We have now added more clarification in the discussion of this result. We tried moving this portion farther down in the section but felt that the order was important to maintain congruency with how the results are presented and to help lead the reader into the discussion about somatisation of mental illness in LMICs.

The aim of this comparative cut-off score analysis was to investigate how various cut-offs scores validated in other perinatal contexts would compare in our sample. As has been shown in similar work around the world the two scales generally perform quite differently within the same context and the same scale performs quite differently across two different contexts. It may be that the EPDS significantly underestimates the incidence of CMD symptoms in this Gambian sample due to how it is administered, the response format and cultural differences in understandings a presentation of perinatal CMD symptoms as has been found in other similar perinatal populations (e.g.,[30,31]). It may also be that the EPDS does not capture the somatic way in which CMD symptoms are experienced. However, without a clinical interview to compare against, it is difficult to know the reason for the observed differences.

(Discussion, Page 16)

8. *Discussion - Personally I think you're likely to be right that the EPDS is under-estimating CMD symptoms. However, in the absence of a gold standard comparison I would be wary about making this assumption so explicitly. I would suggest re-word so that it's suggested as a possibility, whilst being clear that without a clinical interview to compare against it's not possible to know whether the EPDS or the SRQ is providing the more accurate measure.*

We agree that a more nuanced discussion around this finding is needed. We have now added some clarification when talking about this result.

It may be that the EPDS significantly underestimates the incidence of CMD symptoms in this Gambian sample due to how it is administered, the response format and cultural differences in understandings a presentation of perinatal CMD symptoms as has been found in other similar perinatal populations (e.g.,[30,31]). It may also be that the EPDS does not capture the somatic way in

which CMD symptoms are experienced. However, without a clinical interview to compare against, it is difficult to know the reason for the observed differences.

(Discussion, Page 16)

9. *I wondered whether you considered the differences between the two languages in the Gambian setting. Are the two languages very similar to each other? Purists would argue that each language (even if just a dialect of each other) should be validated separately, and different cut-offs might potentially apply to each language. I don't feel that you need to change anything in your analyses, but I think it warrants mentioning in the Discussion section that ideally each of the languages would have been assessed separately.*

We agree and there are differences between the two languages. We have now added a few sentences talking about the differences in language between Mandinka and Wolof and mention how differences in these languages would ideally be assessed separately for a validation study.

For the purposes of this comparative analysis, we combined the scores from the Mandinka and Wolof languages. Mandinka is a Mande language and Wolof is classified as a Niger-Congo language and there is a mutual influence between them. If validation were to be done in the future, ideally each of the languages would be assessed separately.

(Limitations, Page 19)

Reviewer: 2

Dr. M Rahman, University of Malaysia Sarawak

1. *Delete "It is important to be able to detect symptoms of common mental disorders (CMDs) in pregnant women. However, the expression of these disorders can differ across cultures and depend on the specific scale use" from abstract*

We are not quite sure why you have suggested we delete this sentence. We feel it is helpful context for the wide range of readers we hope this paper attracts in BMJ Open so have left this in.

2. *Delete "by examining score distributions and the proportion of women with high levels of symptoms" in abstract*

Again, we are not quite sure why you have suggested we delete this sentence. We feel it is helpful description of our analysis and therefore have left this in.

3. *This study seems that the authors tried to validate the questionnaire from cross-cultural point of view. I did not find anything about content validity of the instruments. How many persons involved in the validation of the instruments? Where is item analysis? Just write the mean the standard deviation is not enough.*

Thank you for your comment. We realise we could be clearer throughout the paper that this study is not about validating the questionnaire. To validate the screening tools, you are right that we would do different types of analyses as well as compare our results to a clinical interview. For this study we have taken a descriptive and comparative approach for our analysis to look at the differences and similarities between Gambian EPDS and SRQ-20 scores as well as Gambian and UK EPDS scores. Through this approach we can contribute some understanding to how CMDs present in pregnant women in The Gambia and compliment other research that has suggested the importance of careful consideration of culture when choosing screening instruments.

We have made this clearer in the methods section by adding these sentences:

This study is a cross-sectional correlation comparison study where we use a comparative and descriptive analysis approach. This is not a validation study but rather exploratory and descriptive in nature to compare how two different scales used to measure perinatal CMD symptoms perform within The Gambia and, also, how the same scale (EPDS) performs across two different countries.

(Methods, Page 5)

- 4. Just describing the items in terms of mean and standard deviation is not enough. It did not make sense about the item reliability. This is the main weakness of this article.*

For a validation study, you are correct that this would not be sufficient. Please see our comment above in relation to this point.

- 5. Exploratory Factor Analysis: Not really convincing results. A substantial sample and analysis would be important to make decision on cross-cultural issues. Results showing low loading.*

We found the exploratory factor analysis an interesting addition to the paper, but we agree with what you and reviewer 3 have suggested and therefore have decided to take this analysis out. We feel the other included analyses are more convincing and there is already enough without this analysis to make an important contribution.

- 6. Factor structures: High factor correlation? Where is other factor correlation. I think it has poor factor discrimination and not convincing results.*

We have now taken out this analysis. See comment above.

- 7. Table sup 4: Why every bracket use % sign?*

We have now taken out the % in each bracket for better readability.

Reviewer: 3

Dr. Paola Rucci, Alma Mater Studiorum – University of Bologna

Comments to the Author:

The paper by Sanfilippo et al. aims at comparing the expression of antenatal symptoms of Common Mental Disorders between pregnant women in Gambia and the UK. Women from Gambia completed both the SRQ-20 (Self Reporting Questionnaire) and the EPDS (Edinburgh Postnatal Depression Scale), while women from the UK completed only the EPDS.

As a general comment, the paper is interesting and helps to underline the importance of investigating differences in antenatal symptoms within and across different cultures. However, there are some methodological issues to clarify and the results need to be improved.

- 1. In particular, the EPDS is an established and validated scale. I would be inclined to suggest to limit the analyses to the comparisons of the total score, the subscale (anxiety and depression scores) and the items between samples, in line with the aims of the study. Exploratory factor analysis in the Gambian sample leads to an unstable solution with three factors and one item with a cross-loading and is not a useful add-on to the paper.*

Thank you for your suggestion. We agree that the factor analysis distracts from the main aims and results of the paper and have decided to take this analysis out.

- 2. The Bland-Altman analysis comparing the EPDS and the SRQ-20 scales in the Gambian sample is not appropriate because the two scales have a different content. A correlation between the two scales would offer more useful information to the readers.*

Thank you for your suggestion. We now just report a Spearman's correlation test. We have also added Supplementary Material 3 which is a scatterplot between EPDS and SRQ-20 total scores.

To investigate differences and similarities in the EPDS and the SRQ-20 within the Gambian sample, a Spearman's correlation test was conducted, differences in the distributional properties were examined and the average score of the EPDS-3 subscale, and its contribution to the total EPDS score of the sample, was computed.

(Statistical Analysis, Page 8-9)

The overall total scores for both scales were significantly moderately correlated $r_s = 0.6$, $p < 0.001$ (See Supplementary Material 3).

(Results, Comparing the EPDS and the SRQ-20 in The Gambian Sample, Page 12)

- 3. Page 3 of 46, Line 43. Authors should report mean EPDS scores along with 95% CIs.*

These have now been added to the Abstract

Gambian participants' EPDS and SRQ-20 scores were significantly moderately correlated ($r_s = 0.6$, $p < 0.00$), had different distributions, 54% overall agreement, and different proportions of women identified as having high levels of symptoms (SRQ-20 = 42% versus EPDS = 5% using highest cut-off score). UK participants had higher EPDS scores ($M = 6.5$, 95%CI[6.1, 6.9]) than the Gambian participants ($M = 4.4$, 95% CI[3.9, 4.9]) ($p < 0.001$, 95%CIs [-3.0, -1.0], Cliff's delta = - 0.3).

(Abstract, Page 2-3)

4. Page 7 of 46

- a. *Line 125, 126. The items that comprise the anxiety subscale are reported inconsistently in the paper. They mention items 1,2, 4 and subsequently (line 246) they mention items 3, 4 and 5. Please check.*

Thank you for catching this. This has now been corrected.

The three anxiety items (items 3, 4 and 5) have been shown to form a valid anxiety subscale (EPDS-3 [22]).

(Measurement Scales, Page 7)

- b. *Line 132-134. The items comprising somatic symptoms and those comprising psychological symptoms should be listed in the text and in table 2.*

This has been added in the text and table 2 updated.

The SRQ-20 includes items measuring common somatic symptoms associated with anxiety and depression (headaches, low appetite, poor digestion, and sleep problems) (items 1–3, 5, 7, 18–20) as well as other psychological and physical/somatic symptoms (feeling frightened, unhappy, worthless, and low-energy) (items 4, 6, 8, 9–17) [8,10]. To each item, participants answer either yes = 1 or no = 0. The scores range from 0 – 20 with a higher score indicating higher levels of CMD symptoms.

(Measurement Scales, Page 7)

5. *Page 8 of 46, Line 156. In 'Statistical analysis', the statistical tests used to compare the demographic and pregnancy characteristics and the item frequencies are not indicated. The tests used and the corresponding significance level should also be reported in Table 1*

What tests were used have now been added into the Statistical Analysis section

To compare the demographic and pregnancy characteristics independent two-sample t-tests were used.

(Statistical Analysis, Page 8)

Individual item frequencies on the SRQ-20 (# of yes responses out of total responses) were also calculated.

(Statistical Analysis, Page 9)

The average score of each item and of the EPDS-3 subscale was calculated. Its contribution to the total EPDS score of the sample was also computed. Differences between the total EPDS score between the UK and The Gambia were calculated using a Mann-Whitney U test. (Statistical Analysis, Page 9)

We have added the below to the Note section of Table 1 to indicate the tests that were done and the significance levels.

*Indicates a significant difference ($p < 0.05$) between the groups using independent samples t-tests).

(Table 1, Page 12)

6. *Page 9 of 46, Line 171, 172. Please remove the sentence "The average scorecomputed" because it is a repetition of the same sentence reported in lines 159, 160.*

This has been removed.

7. *Page 10 of 46*

- a. *Line 195. In Table 1, the absolute and percentage frequencies should be reported for the total samples of each country, indicating the missing data for each variable. Apparently for UK data, demographic information is available only for 314 women.*

The absolute and percentage frequencies are included in Table 1. The missing data has now been clarified in the Table and in the Note section of Table 1.

**The total sample in the previously conducted UK study was 368 but there was missing demographic data (about 54 participants). The total used for each category is included.

(Table 1, Page 12)

b. Line 195. In Table 1, a parenthesis is missing for GA's standard deviation.

This has now been added.

c. Line 195. Please round off percentages, means and standard deviations to one decimal figure throughout the text and tables.

This has now been done throughout that paper.

8. Page 12 of 46

a. Line 207-209. Given the different content of the scales, the Bland-Altman analysis is useless. I would suggest reporting only the Spearman's correlation coefficient between the EPDS and the SRQ-20.

Thank you for this suggestion. We have now reported a Spearman's correlation instead.

The overall total scores for both scales were significantly moderately correlated $r_s = 0.6$, $p < 0.001$ (See Supplementary Material 3).

(Results, Comparing the EPDS and the SRQ-20 in The Gambian Sample, Page 12)

b. Line 218-221. The authors do not report which and how many items comprise the subscales. However, it seems that subscales consist of a different number of items. If this is the case, comparing the means of subscales is not useful. The authors should consider standardizing the scores before comparison.

Thank you for catching this. We decided we do not want to compare the sub-scales to each other but rather show what the relative contribution to the total score is. We have now made this clearer in how we report these results.

The somatic symptom items contributed 63% to the total SRQ-20 score. The psychological/cognitive/functional symptom items contributed 37%.

(Results, Comparing the EPDS and the SRQ-20 in The Gambian Sample, Page 13)

9. *Page 13 of 46, Lines 226-230. Please use Mann-Whitney test to compare mean EPDS scores between countries, given the skewness in the Gambian sample.*

Thank you for this suggestion. We have now amended the statistical analysis section and results section to reflect this change.

Differences between the total EPDS score between the UK and The Gambia were calculated using a Mann-Whitney U test.

(Statistical Analysis, Page 9)

The UK average EPDS score (M = 6.5, SD = 4.4, 95%CI[6.1, 6.9]) was significantly higher than the Gambian average score (M = 4.4, SD = 3.7, 95% CI[3.9, 4.9] ($p < 0.001$, 95%CI [-3.0, -1.0], Cliff's delta = -0.3) ($p < 0.001$ 95%CI [1.5, 2.8], $d = 0.5$). A Mann-Whitney U Test was used because the UK scores were slightly positively skewed (skewness = 0.63), while the Gambian EPDS scores were highly positively skewed (skewness = 1.37), reflecting a higher frequency of lower scores in the Gambian sample.

(Comparing EPDS scores in The Gambia and the UK, Page 14)

10. *Page 14 of 46*

- a. *Line 234. I would suggest comparing the proportion exceeding the 3 cut-off scores in the two countries using chi-square test and reporting the results in a table.*

Thank you for this suggestion. Reviewer 1 pointed out that the analysis focused on cut-off scores needed some more careful wording around some of its limitations due to the fact that the cut-off scores used are not validated against clinical interview. She also recommended we focus less on these results in our discussion. Because of this, we have reworked the explanation of these findings and feel that an added analysis as you suggest might not be needed as there are other elements of the paper we hope to focus more on.

You can see the updated wording around the discussion of these results below.

When comparing scores on the EPDS between the Gambian and UK samples, the UK participants had significantly higher total EPDS scores and 13% of UK participant's scores were met threshold for high levels of symptoms (≥ 12) compared with only 5% of the Gambian sample. This could, of course, be taken at face value to indicate potentially higher levels of CMS symptoms in the UK population. However, the comparison of EPDS and SRQ-20 scores reveals an alternative view. While both scales were significantly moderately correlated ($r_s = 0.6$, $p < 0.00$), they had different distributions and only 54% overall agreement. Additionally, when using previously validated cut-off

scores from other contexts, only 5% of Gambian participants met threshold for high levels of symptoms with the EPDS, while that number rose to 42% when the SRQ-20 was used.

The aim of this comparative cut-off score analysis was to investigate how various cut-offs scores validated in other perinatal contexts would compare in our sample. As has been shown in similar work around the world the two scales generally perform quite differently within the same context and the same scale performs quite differently across two different contexts. It may be that the EPDS significantly underestimates the incidence of CMD symptoms in this Gambian sample due to how it is administered, the response format and cultural differences in understandings a presentation of perinatal CMD symptoms as has been found in other similar perinatal populations (e.g.,[30,31]). It may also be that the EPDS does not capture the somatic way in which CMD symptoms are experienced. However, without a clinical interview to compare against, it is difficult to know the reason for the observed differences.

(Discussion, Page 16)

- b. *Line 249. The percentage of contribution of the anxiety subscale to the total UK EPDS score is reported as 60%. However, if we divide 3.49 (the average scores of items #3, #4 and #5) by 6.51 (the total score), the percentage should be 54%. Please check.*

Thank you for catching this mistake. This has now been amended.

- c. *Line 254. Please report in the table 3 the subscale average scores.*

These has now been added to Table 3.

11. *Page 15 of 46*

- a. *Line 256. The Bonferroni correction to the probability level applied to 10 items should be 0.05/10 and not 0.05/13. Please revise.*

You are right that there are 10 items on the scale, but we also are adjusting for comparisons we did for each sub-scale (2) and the total score (1). That makes a total of 13 comparisons. Therefore, this is why the Bonferroni correction to the probability level was calculated using 0.05/13.

- b. *Line 264-266. The variance explained by each factor should be reported at the beginning of table 4 for the two countries.*

The factor analysis has now been taken out. You can see why in our response to Reviewer 2.

12. *Page 23, Please replace ref [19] with Kabir K, Sheeder J, Kelly LS. Identifying postpartum depression: are 3 questions as good as 10? Pediatrics. 2008 Sep;122(3):e696-702, where the three items comprising the anxiety scale are listed.*

We have now added Kabir et al to the references for this sentence.

13. Page 27 of 46 Figure 2. Authors should specify in the x-axis of the histogram "Gambian EPDS overall score"

This has been added.

Reviewer: 4

Dr. Susan Pawlby, Institute of Psychiatry, Psychology & Neuroscience at King's College London

1. *This manuscript is extremely well written and addresses some of problems faced in identifying common mental health problems (CMDs) in Lower and Middle Income Countries when using screening instruments developed in English for use in Higher Income Countries. In this study the authors firstly compared Gambian pregnant women's responses to the Edinburgh Postnatal Depression Scale (EPDS), developed in the UK, and the Self-reporting Questionnaire (SRQ-20) developed by the WHO for use across cultures. Secondly they compared responses to the EPDS in pregnant women in The Gambia and UK. Both instruments (EPDS and SRQ-20), were rigorously translated, back-translated and given orally to the Gambian women, with each tool contributing to the understanding of how CMDs present in pregnant women in The Gambia. As in other African countries there was more endorsement of somatic symptoms, suggesting that the SRQ might work better in identifying CMDs among pregnant women in these cultures than the EPDS. Comparison of the Gambian women's scores on the EPDS with those from a group of UK women confirmed that women in the UK scored significantly higher than the Gambian women and that there was a different factor structure. Although no structured diagnostic interview was given to the Gambian women (a limitation acknowledged by the authors), other African studies have shown good criterion validity of the SRQ against the SCID and the MINI. This study in The Gambia is important as it complements other African studies examining ways in which CMDs can be assessed in antenatal clinics so that pregnant women can be supported through the perinatal period. The reliance on screening instruments and indeed interviews that have originally been formulated in Western countries and often in the English language, some of them many years ago, still poses a problem, despite the extremely rigorous attention that has been paid to translation, using consensus rating and focus groups. Few studies have used instruments developed specifically for use in a given cultural setting. More qualitative research conducted within country by local health professionals with pregnant women should help characterise perinatal CMDs in different African contexts.*
2. *Following the concluding sentence that 'This study helps to further underline the importance of investigating differences in scale performance within and across cultures to ensure that the measurement scale chosen is sensitive to cultural differences in the understanding and presentation of perinatal common mental disorders', the authors may like to refer to a recent study by Bluett-Duncan who presented another way of overcoming the problems of crosscultural research while using multi-item scales of mental health. He explored Differential Item Functioning (DIF) comparing a group of Indian mothers with a group of UK mothers in their responses to the items of the EPDS. In his thesis he describes the development of a set of anchoring vignettes (AVs) in order to assess the impact of DIF on the EPDS. His findings showed that the mothers in the two cultural settings rated the items on the EPDS very differently (as they did in The Gambia compared with the UK). Prior to Differential Item Functioning (DIF) adjustment, rates of depression at 12 months were significantly higher in UK participants, compared with Indian participants, but after DIF adjustment, rates were significantly higher in Indian participants. Differences in response style indicated that Indian women were under-reporting depressive symptoms when compared with the UK women and*

showed that scores unadjusted for DIF should not be taken at face value as they may mask the extent of maternal mental ill-health in LMICs. Bluett-Duncan, M. (2021). A cross-cultural comparison of the role of maternal mental health in the prediction of infant cognitive development and empirical investigation of the role of early caregiving in India (Doctoral dissertation, University of Liverpool).

Thank you for your kind and thorough review. We have found your words around the importance of this study very helpful and clear. This has inspired some minor changes to how we discuss the impact of this work. You can see our changes below:

While careful consideration and translation of the SRQ-20 and EPDS helped to ensure semantic validity of the scales for use in The Gambia, the scales have not been validated against a clinical gold-standard (clinical interview) to screen for possible CMD, highlighting a limitation of this work. Other studies conducted in sub-Saharan African countries with perinatal populations studies have shown good criterion validity of the SRQ-20 and the EPDS against the clinical interview [10,25,38–40]. However, validation against a clinical interview was difficult in The Gambia partly because there are few trained professionals (only two trained psychiatrists for the whole of the country) able to conduct these interviews in the two local languages. This limitation highlights an important consideration needed when conducting validation studies in low resource settings with few local trained health professionals. Even though the scales were not validated we feel the comparison of the scales' performances when using possible cut-off scores is an interesting and helpful exploratory process.

(Limitations, Page 18)

Crucially, this study helps to further underline the importance of investigating differences in scale performance within and across cultures to ensure that the measurement scale chosen is sensitive to cultural differences in the understanding and presentation of perinatal common mental disorders. Without careful exploration of these differences, research in this area may have detrimental consequences in detection and therefore treatment of perinatal mental health in LMICs.

(Conclusion, Page 20)

Also, we found Bluett-Duncan's work very relevant to our work and have now referred to this work in the discussion:

It may be that the EPDS significantly underestimates the incidence of CMD symptoms in this Gambian sample due to how it is administered, the response format and cultural differences in understandings a presentation of perinatal CMD symptoms as has been found in other similar perinatal populations (e.g.,[30,31]).

(Discussion, Page 16)